# Analytical Validation of a Direct Competitive ELISA for Multiple Mycotoxin Detection in Human Serum

**DOI:** 10.3390/toxins14110727

**Published:** 2022-10-25

**Authors:** Kunal Garg, Fausto Villavicencio-Aguilar, Flora Solano-Rivera, Leona Gilbert

**Affiliations:** 1Tezted Ltd., Mattilaniemi 6-8, 40100 Jyväskylä, Finland; 2Sanoviv Medical Institute, KM 39 Carretera Libre Tijuana-Ensenada s/n Interior 6, Playas de Rosarito, Baja 11 California, Rosarito 22710, Mexico

**Keywords:** human biological monitoring, multiple mycotoxins, enzyme-linked immunosorbent assay

## Abstract

Mycotoxin exposure in humans is primarily assessed through its occurrence in external sources, such as food commodities. Herein, we have developed a direct competitive ELISA to facilitate the detection of aflatoxin B1 (AFB1), deoxynivalenol (DON), fumonisin (FUM B1/B2), ochratoxin A (OTA), and zearalenone (ZEA) in human serum. The analytical validation of the assay followed practices endorsed by the international research community and the EU directive 96/23/EC in order to examine detection capability, recovery, and cross-reactivity. The assay demonstrated a lower limit of quantitation (LLOQ) for AFB1 [0.61 ng/mL (hereon ng/mL = ppb)], DON (19.53 ppb), FUM (4.88 ppb), OTA (19.53 ppb), and ZEA (0.15 ppb). Recovery from human serum for all mycotoxins spanned from 73% to 106%. Likewise, the specificity for monoclonal antibodies against cross-reactant mycotoxins ranged from 2% to 11%. This study compares the LLOQ and recovery values with commercial and emerging immuno-based methods for detecting mycotoxins in foodstuffs. The LLOQ values from the present study were among the lowest in commercial or emerging methods. Despite the differences in the extraction protocols and matrices, the recovery range in this study, commercial tests, and other procedures were similar for all mycotoxins. Overall, the assay detected AFB1, DON, FUM, OTA, and ZEA in human serum with excellent accuracy, precision, and specificity.

## 1. Introduction

Many fungal species like Penicillium, Fusarium, Aspergillus, and Alternaria produce secondary metabolites known as mycotoxins [1]. Mycotoxins are produced by mold in various foodstuffs [2]. Thus, several guidelines by the EU commission (decision no. 2002/657/EC [3] or regulation no. 401/2006 [4]), Food and Drug Administration (FDA) [5], or the European Medicines Agency [6] mandate mycotoxin detection in crops, animals, and animal products in order to reduce their public health risk. In 1985, the Food and Agriculture Organization (FAO) estimated that up to 25% of all crops contained mycotoxin contamination [7]. Today, the ever-improving sensitivity of analytical methods indicates that 60% to 80% of crops have mycotoxin contamination [7]. Advanced food processing technologies cannot eliminate all mycotoxin compounds [8].

Human biological monitoring (HBM) for mycotoxins is becoming essential due to the presence of mycotoxins in our food [9] and exposure through the skin or the inhalation of mold/spores [10]. Mycotoxins in humans at low or chronic levels of exposure are related to hepatotoxicity, carcinogenicity, endocrine disorders, and nephrotoxicity [11]. The damage caused by mycotoxins depends on their type, amount, and the duration of exposure, among other factors [12]. Aflatoxin B1 (AFB1), deoxynivalenol (DON), fumonisin (FUM), ochratoxin A (OTA), and zearalenone (ZEA) are of common interest in foodstuffs [13]. In humans, AFB1 adducts disrupt the DNA repair mechanism [14], DON may bind to ribosomal subunits and inhibit protein synthesis [15,16], FUM promotes a pro-inflammatory environment [17], OTA causes renal tumors [18], and ZEA leads to hormonal imbalances in women [19].

Climate change will present favorable growing conditions for fungi to promote AFB1 and DON contamination in maize and wheat in Europe [20]. Thus, monitoring crops for mycotoxin contamination will remain essential for controlling toxin-related disease outbreaks. However, correlating human exposure to mycotoxins solely based on data from crop contamination manifests three challenges:The heterogeneous distribution of mycotoxin in food and the matrix effect during analytical detection can result in an underestimated exposure level [21,22].Accurate data on food consumption and source for every individual is not always available to trace the origin and levels of exposure [21,22].Mycotoxin in food does not always imply exposure as their bioavailability, food treatment or composition, and individual-to-individual differences are diverse [8].

Individual HBM can help to circumvent the abovementioned challenges by directly measuring mycotoxins in human matrices [23,24]. Additionally, HBM can help to identify mycotoxins in individuals who lack exposure information [23,24]. Between 2015 and 2020, 61% of the literature on HBM employed liquid chromatography (LC) with tandem mass spectrometry (LC/MS/MS) to measure up to 19 mycotoxins in human plasma [10]. LC/MS/MS is also the gold standard for the detection of multiple mycotoxins in foodstuff; however, an LC/MS/MS is expensive, needs highly trained professionals for operation, and its sensitivity depends on the type of ionization technique [10]. Alternatively, gas chromatography with mass spectrometry (GC/MS) can achieve multi-mycotoxin detection, but GC/MS is inefficient because it requires the derivatization of mycotoxins for detection and suffers from poor reproducibility [10]. Liquid chromatography with a fluorescence detector (LC-FLD) provides accurate and reproducible multi-mycotoxin detection at a considerably lower price and requires fewer resources than LC/MS/MS or GC/MS. Nevertheless, LC-FLD cannot detect non-fluorescent mycotoxins, such as DON and T-2/HT2 [10].

LC/MS/MS can only aid in directly determining unmodified mycotoxins that are available as reference substances and indirectly detecting metabolized mycotoxins that have been reduced to their free form [10]. For the non-targeted analysis, researchers have developed LC with high-resolution MS (LC-HRMS) in order to detect unknown mycotoxin derivatives in human biofluids [25]. Furthermore, HBM research requires an easily accessible matrix for sampling and analyzing mycotoxins. For example, human urine contains all mycotoxin biomarkers, and its collection process is non-invasive [26]. Nevertheless, the variability of mycotoxin concentration in urine and its volume based on daily intake demands urine sampling at different time points during the day and the normalization of results with creatinine concentration [27]. The inter-individual comparison of mycotoxins in urine is challenging because various factors, such as gender, age, diet, and muscle mass, can influence creatinine secretion [27].

Herein, we propose using the direct enzyme-linked immunosorbent assay (ELISA) format to produce an accurate, sensitive, and selective multi-mycotoxin detection tool for human serum. The present study is critical because LC machines are expensive and not easily accessible [10] in universities, clinical laboratories, and hospitals. Hence, an affordable and sustainable approach for measuring mycotoxins in human matrices remains an unmet need. Additionally, unlike human urine, serum or plasma contains higher levels of mycotoxins, and the results indicate long-term exposure [10].

## 2. Results

The results presented below provide an analytical proof-of-concept for using the direct competitive ELISA format for measuring AFB1, DON, FUM, OTA, and ZEA in human serum. Figure 1a,b establishes calibration curves with a four-parameter logistic (4PL) and Gaussian fit for AFB1, DON, FUM, OTA, and ZEA. The 4PL fit coefficient of determination (R²) values ranged between 0.97 and 0.99 for AFB1, DON, FUM, OTA, and ZEA, indicating a near-perfect fit for all mycotoxins (Figure 1a). Similarly, the R² values ranged from 0.89 to 0.99 for the Gaussian fit (Figure 1b). Figure 1a also reveals the LLOQ for AFB1 [(0.61 ng/mL (hereon ng/mL = ppb)], DON (19.53 ppb), FUM (4.88 ppb), OTA (19.53 ppb), and ZEA (0.15 ppb). The 4PL fit accuracy at the LLOQ ranged between 93% and 111% for AFB1, DON, FUM, OTA, and ZEA (panel a). The calibration curves in Figure 1 validate the assay’s ability to accurately measure low amounts of mycotoxin.

The log10 concentrations were plotted on a logarithmic x-axis in Figure 1 in order to obtain a 4PL fit; otherwise, a linear x-axis with negative log10 values would cross over the y-axis and change the behavior of the 4PL curve. The Gaussian curves in Figure 1b were made for back calculating the log10 concentration for the recovery and matrix interference analyses. Following the extraction of mycotoxins from spiked human sera, the mean recovery for all mycotoxins spanned from 73% to 106% (Table 1). An increase in the spike concentration improved the recovery percentage across mycotoxins for FUM, OTA, and ZEA, but also increased variation [coefficient of variation (%CV)] due to fluctuations in optical density (OD) values at the upper limit of detection (Figure 1 and Table 1). It is important to note that the non-spiked serum that was pre-treated and extracted with 1% formic acid in acetonitrile provided similar optical density (OD) values to non-spiked 2% BSA (Appendix A). The variation (%CV) between non-spiked serum and 2% BSA was less than 10% and suggested that serum from the healthy donor was negative for AFB1, DON, FUM, OTA, and ZEA (Appendix A).

The recovery percentages in Table 1 demonstrate that the extraction procedure can successfully isolate the mycotoxins of interest. Over 85% of matrix factor (MF) and matrix effect (%ME) values in Table 1 revealed matrix suppression effects on mycotoxin recovery. The suppression influence was more significant at low spiking concentrations as opposed to high spiking concentrations (Table 1). However, a relationship between the recovery percentages and MF or %ME could not be drawn. For example, ZEA presented higher recovery percentages at all spiked concentrations despite suffering from more significant matrix suppression compared to FUM or OTA (Table 1). The extraction procedure enabled the accurate and consistent quantification of AFB1, DON, FUM, OTA, and ZEA (Table 1). Overall, the assay presented here can measure different mycotoxins at various amounts with high precision (Table 1).

Lastly, the cross-reactivity was 100% between the antibodies and their matching pair of mycotoxin standards (Figure 2), demonstrating that Equation (5) works as expected. Figure 2 also illustrates (in a heatmap format) that the cross-reactivity percentage varied between 2% and 11% in all other combinations, meaning that the monoclonal antibodies did not present a significant response to cross-reactant mycotoxins.

## 3. Discussion

The risk of mycotoxin exposure in humans is predominantly assessed through its occurrence in external sources, such as food commodities [28,29]. However, mycotoxins are not homogenously distributed across diverse matrices, and their bioavailability can vary in individuals due to various food treatment and consumption patterns [8,21,22]. Thus, the direct determination of mycotoxins in humans is crucial for understanding their prevalence and pathogenesis in diseases [10]. LC/MS/MS is the method of choice for monitoring mycotoxins in humans [10], but the approach requires expensive machinery and highly trained professionals. Herein, we have developed a direct competitive ELISA to facilitate the detection of AFB1, DON, FUM B1/B2, OTA, and ZEA in human serum (Figure 1 and Figure 2, Table 1).

Guidelines for validating analytical methods that aim to detect mycotoxins in human body fluids do not exist. We have adapted analytical validation practices endorsed by the international research community [30] and the EU directive 96/23/EC [3] in order to examine the calibration curves, detection capability, recovery, repeatability, matrix interference, and cross-reactivity for five different mycotoxins in our multiplex ELISA. Our assay measured low amounts of mycotoxins with high accuracy and near perfect 4PL or Gaussian fits (Figure 1). The extraction procedure recovered the mycotoxins despite matrix interference (Table 1). Furthermore, our assay measured isolated mycotoxins with high precision (Table 1). We noted considerable specificity for all monoclonal antibodies in Figure 2. Overall, the assay can aid in detecting AFB1, DON, FUM B1/B2, OTA, and ZEA in human serum with significant accuracy, precision, and specificity.

The monitoring of mycotoxins in humans through ELISA is limited to AFB1 adducts with albumin or lysine [31,32]. As a result, it is challenging to formulate a head-to-head comparison of all the analytical characteristics in Figure 1 and Figure 2 and Table 1. Immunoassays are typically utilized in academia and industry for studying mycotoxins in foodstuffs [33,34,35,36,37,38]. Moreover, the principles of direct competitive ELISA [39] stay immutable regardless of the diverse matrices and their influence on the assay (i.e., food, feed, or human body fluid). This study compares LLOQ (Table 2) and recovery (Table 3) with commercial and emerging immuno-based methods for detecting mycotoxins in foodstuffs [33,34,35,36,37,38]. We pulled together the analytical performances of commercial assays from Cusabio [38], Elabscience [40,41,42,43,44], Helica^TM^ [45,46,47,48,49], AgraQuant^®^ [50,51,52,53,54], and VICAM [55,56,57,58,59] in Table 2 and Table 3. Unlike commercial tests, modern methods [33,34,35,36,37], such as lateral flow immunoassay (LFIA) or surface plasmon resonance (SPR), focus on detecting multiple mycotoxins (Table 2 and Table 3).

Table 2 also examines the LLOQ values from Figure 1 against the EU’s lowest guidance levels for mycotoxins in foodstuffs [60]. Our LLOQ values are 3 to 333 times lower than the EU guidance level for AFB1, DON, FUM, and ZEA (Table 2). For the same mycotoxins, our LLOQ values were at least 2.5- to 40-fold less than the smallest quantitation or detection limit for commercial mycotoxin tests (Table 2). Similarly, LFIA and SPR methods also demonstrate multi-mycotoxin detection [33,34,35,36,37], with quantitation or detection limits that were 5- to 40-fold lower than commercial tests (Table 2). The LLOQ values from the present study were amongst the lowest values observed in commercial and emerging immuno-based methods for AFB1, DON, FUM, and ZEA (Table 2). Other techniques, such as chemiluminescence (CLIA) and surface-enhanced Raman scattering (SERS), can further reduce the detection limits to 0.0001 ppb for AFB1, DON, or OTA [61,62]. However, due to high background signal or low signal, selectivity, accuracy, and precision can suffer in CLIA and SERS methods.

The recovery percentages were unavailable or not indicated for the Cusabio [38], AgraQuant^®^ [50,51,52,53,54], VICAM [55,56,57,58,59], Xing et al. [34], and Joshi et al. [36] assays demonstrated in Table 2. Thus, Table 3 compares the recovery percentages from the remaining sources in Table 2 and the current study (Table 1). We employed acetonitrile with 1% formic acid to extract mycotoxins from human serum, whereas the commercial tests and emerging methods in Table 3 used 20% to 90% methanol for the solid–liquid extraction of mycotoxins from foodstuffs [33,35,37,40,41,42,43,44,45,46,47,48,49]. Despite the differences in the extraction protocols and matrices, the recovery range in our study (73–106%), commercial tests (70–119%), and other methods (82–123%) were similar for AFB1, DON, FUM, OTA, and ZEA (Table 3). We observed that the upper recovery percentages in other studies [33,35,37,40,41,42,43,44,45,46,47,48,49] often exceeded 100% for all mycotoxins compared to the values from our study (Table 3). Information regarding matrix interference was absent from all the comparative studies in Table 3. We postulated that matrix enhancement might account for unusually high recovery percentages as opposed to matrix suppression in this study (Table 1).

In Table 2, OTA exhibits an LLOQ value (19.53 ppb) that is higher than the EU guidance level (2 ppb) and the quantitation or detection limits of other studies (0.2–13 ppb), suggesting the potential for further improvements in our assay. Traditional comparisons of mycotoxin assay performance include index and reference tests using the same food matrices, such as wheat, barley, or corn [33,34,36,37,42,44,45]. However, Table 2 and Table 3 anchor our assay performance with commercial tests and emerging techniques that have been optimized to detect mycotoxins in food or feed matrices (Appendix A) because we lack the precedence for similar immuno-recognition tools using human biofluids. It is important to note that the assay performance comparison in Table 2 and Table 3 is not a head-to-head comparison of detection capability and recovery percentages, but rather a comparison between our assay, which shares standard features with the other tests (Table 2 and Table 3), such as establishing direct detection and non-linear calibration curves with unmodified reference standards for all mycotoxins [33,34,35,36,37,42,43,44,45].

In Figure 1 and Figure 2 and Table 1, we provide an analytical proof-of-concept for using the direct competitive ELISA format for detecting AFB1, DON, FUM, OTA, and ZEA in human serum with significant accuracy, precision, and specificity. The study used reference material for all mycotoxins and, as is made evident in Appendix A, serum from the healthy donor was negative for AFB1, DON, FUM, OTA, and ZEA. In future studies, we aim to develop an LC method that can confirm and complement the results from the ELISA, incorporate mycotoxins that are modified and structurally homologous to AFB1, DON, FUM, OTA, or ZEA, and clinically validate the assay using positive and negative reference sera specimens. We seek to examine the assay for ruggedness and stability following the Youden approach [30] and guidance from the clinical and laboratory standards institute [63,64], respectively.

## 4. Conclusions

LC/MS/MS, which is a prerequisite for detecting multiple mycotoxins in humans [10], will impede the creation of new knowledge because it is not commonly used in universities, clinical laboratories, and hospitals. We have developed an ELISA method to detect AFB1, DON, FUM B1/B2, OTA, and ZEA in human serum with significant accuracy, precision, and specificity. Emerging techniques [33,34,35,36,37,42,43,44,45], such as LFIA, SPR, CLIA, and SERS, make ELISA seem like an old-fashioned method. Still, ELISA is routinely applied in academia and in industry to diagnose different diseases [65,66,67], which allows our assay to be used without needing new infrastructure, in contrast with SPR, CLIA, and SERS, which demand special machinery. Human biomonitoring may be easier for researchers if they adopt a two-tier approach involving a laboratory-developed ELISA and outsourcing LC for screening and confirming mycotoxins [68].

## 5. Materials and Methods

### 5.1. Reagents and Equipment

AFB1, DON, FUM B1, FUM B2, OTA, and ZEA reference materials were purchased from LGC standards GmBH, Germany, with a certificate of analysis confirming the purity of said mycotoxins using liquid chromatography with ultraviolet, fluorescence, or diode-array detectors. Nunc MaxiSorp™ ELISA plates with a plate shaker, recombinant protein G, methanol, acetonitrile, formic acid, and the Guardian™ peroxidase conjugate diluent were procured from Fisher Scientific, Vantaa, Finland. At 20 mg/mL concentration, FUM B1 and B2 were stored in 50% acetonitrile:water at 4 °C, and the remaining mycotoxins were stored in 100% methanol at −20 °C. Monoclonal antibodies against AFB1, DON, FUM B1/B2, OTA, ZEA, and their conjugates with horseradish peroxidase (HRP) were obtained for direct competitive ELISA from Creative Diagnostics, USA.

SeramunBlock NP and 3,3′,5,5′ tetramethylbenzidine substrate (TMB) were acquired from Seramun GmBH in Germany and Neogen Corporation in the UK, respectively. Phosphate buffer saline (1 × PBS, pH 7.0 to 7.2) was prepared in-house using sodium chloride (137 mM), potassium phosphate monobasic (2 mM), disodium hydrogen phosphate (8 mM), and potassium chloride (3 mM) from Merck, Finland. To coat the recombinant protein G on the Nunc MaxiSorp™ ELISA plate, 100 mM carbonate buffer (pH 9.5) was prepared from sodium carbonate (Merck, Finland) and bicarbonate (Merck, Finland). Sera from a healthy human donor (Merck, Finland) was used to scrutinize mycotoxin recovery and the matrix effects. The study employed EL406 and Synergy H1 with Gen5 software (BioTek Instruments, Winooski, VT, USA) to wash and read the ELISA microplate.

### 5.2. Development of a Direct Competitive ELISA

The optimal amounts of recombinant protein G, monoclonal antibodies, and HRP conjugated mycotoxins were determined through checkerboard titrations [39]. The ideal titration values for monoclonal antibodies were two (anti-AFB1 and anti-DON), four (anti-OTA and anti-ZEA), or sixteen (anti-FUM) times the HRP labelled mycotoxin quantities. MaxiSorp™ ELISA plates were coated with 100 µL of protein G [2500 ng/mL (hereon ng/mL = ppb)] per well in 100 mM carbonate buffer and were incubated at RT for 1 h. Post incubation, the plates were washed five times with 200 µL 1 × PBS using the EL406 microplate washer and were immediately coated with 100 µL per well of monoclonal antibodies against AFB1 (625 ppb), DON (2500 ppb), FUM (2500 ppb), OTA (625 ppb), and ZEA (1250 ppb) diluted in 1 × PBS [39]. After incubation at RT for 1 h, the plates were washed five times with 200 µL 1 × PBS and were simultaneously blocked/stabilized with 300 µL of SeramunBlock NP. The plates were then allowed to incubate for 1 h at RT. Post incubation, the plates were washed once with 300 µL of double-distilled water, air-dried for 1 h RT, and stored at 4 °C in a sealed foil wrap (Waccomt Pack Inc., Amazon.co.uk) with desiccant (Wisedry, Amazon.co.uk).

### 5.3. Construction of Calibration Curves through Competition

A 2% BSA (Merck, Finland) in 1× PBS functioned as the diluent for all mycotoxin standards and as a surrogate to the human matrix [69,70,71]. Mycotoxin standards were dissolved in 2% BSA/PBS using a four-fold serial dilution which ranged between 1250 and 0.076 ppb (DON), 625 and 0.038 ppb (AFB1 and ZEA), and 312.5 and 0.019 ppb (FUM B1/B2). The Guardian™ reagent (Fisher Scientific, Finland) was used to prepare HRP conjugated AFB1 (312.5 ppb), DON (625 ppb), FUM (156.25 ppb), OTA (156.25 ppb), and ZEA (312.5 ppb). Microwells coated with a monoclonal antibody against AFB1 were first introduced to 50 µL of AFB1 standard in 2% BSA/PBS and then 50 µL of AFB1-HRP in Guardian™ reagent [39]. The same procedure involving the addition of a mycotoxin standard and its HRP conjugate was applied to DON, FUM, OTA, and ZEA. After 1 h of incubation at RT on a plate shaker (Fisher Scientific, Finland, 500 rpm), the plates were washed five times with 200 µL 1 × PBS and were then supplemented with 100 µL of TMB. After 30 min of incubation in the dark at RT, 100 µL of 2 M sulphuric acid stopped the catalytic reaction between TMB- and HRP-labeled mycotoxins [39]. Finally, the plates’ optical density (OD) was measured at 450 nm using the bottom-reading approach with Synergy H1.

### 5.4. Pretreatment Method to Determine Recovery and Matrix Effects

Acetonitrile with 1% formic acid (*v*/*v*, 1.2 mL) was added to 0.4 mL of spiked and blank human sera, vortexed for 10 sec, mixed through shaking at RT for 5 min, and centrifuged at 12,000× *g* 4 °C for 10 min [72,73]. All the supernatant was loaded into the Oasis HLB cartridge (Waters, Finland), and then the vacuum for slow elution was applied 5 min later. After drying the elution at 70 °C, 0.4 mL of 2% BSA/PBS was used to reconstitute the samples [72]. For each mycotoxin, the human sera were spiked at 0, 1, 4, 16, and 64 times the LLOQ (Figure 1a) before (spiked) or after [post-extraction spiked (PES)] extraction. The competitive ELISA procedure described above for the calibration curve was followed after reconstitution in 2% BSA/PBS for recovery and matrix interference assessments. Note that mycotoxin-free reference human serum is not commercially available. As a result, the pretreatment method was also applied to plain sera in duplicate and was compared with 2% BSA to ensure that the healthy donor serum did not interfere with the recovery and matrix effect analyses for each mycotoxin.

### 5.5. Characterizing Assay Specificity

Specificity was characterized for each monoclonal antibody by detecting cross-reactant mycotoxins [74] spiked in 2% BSA/PBS, including AFB1 (625 ppb), DON (1250 ppb), FUM B1/B2 (312.5 ppb), OTA (312.5 ppb), and ZEA (625 ppb) per well [74,75].

### 5.6. Data Analysis

The study includes calibration curve data performed on three different days in duplicates and the use of www.myassay.com (accessed on 29 September 2022) and www.mycurvefit.com (accessed on 29 September 2022) to fit regression curves. The four-parameter logistic (4PL) and Gaussian fits were employed to draw a non-linear relation between the mycotoxin concentrations in 2% BSA/PBS (abscissa, x-axis) and OD at 450 nm (ordinate, y-axis) [30,76,77]. The 4PL and Gaussian fitting process included background correction, log10 transformation of mycotoxin concentrations, and signal-to-noise (%B/B0) assessment [33,34,35]. In particular, %B/B0 refers to the OD value with (B) and without (B0) competition for antibody binding sites on the microplate. Replicated data points were fitted with 4PL so that the curve’s upper and lower asymptotes equal B and B0 (0 and 100%) [30,76,77]. In contrast, the Gaussian fitting only included the average %B/B0 from the replicates on the ordinate scale.

The ‘My Assay’ program was used to obtain the coefficient of determination (R²), the LLOQ, and accuracy percentages at said LLOQs achieved through back-calculation. The coefficient of determination (R²) is the ratio of variation that the 4PL curve-fitting model explains to the total variation in the model [30,78]. The R² value will equal 1 for a perfect fit and move closer to 0 for a bad 4PL fit [30,78]. Furthermore, the lower limit of quantitation (LLOQ) is the lowest amount of an analyte detectable in a sample with reasonable accuracy and precision [30]. While 4PL analysis helped to characterize the R² and LLOQ for all the antibodies, we used coefficients (a, b, and c) from the Gaussian curve to back-calculate log10 concentration (Equation (1)) and examine the mycotoxin recovery percentages (Equation (2)), the matrix effect (ME, Equation (3)), and the matrix factor (MF, Equation (4)).
f(x) = 〖ae〗^ (−〖(x − b)〗^2^/〖2c〗^2^)(1)

In Equation (1), a = curve peak height, e = Euler’s number, x = integer, b = position of the peak’s center, and c = standard deviation.
Recovery% = 100 × (Extracted sample concentration)/(Post extracted spiked sample concentration)(2)
ME% = 100 × (Post extracted spiked sample concentration)/(Sample concentration in surrogate matrix)(3)
MF = (Extracted sample concentration)/(Sample concentration in surrogate matrix)(4)

In Equations (2)–(4), concentration refers to the log10 values predicted using the Gaussian curve and the associated coefficient values in Figure 2 for each mycotoxin.

Experiments for recovery and matrix effect were repeated on two different dates in duplicates for all the mycotoxins. Equation 2 and the coefficient of variation (%CV) were used to identify the accuracy and precision of the assay, which are expressed as average recovery (%) and repeatability (%CV). For matrix interference (Equations (3) and (4)), MF values < 1 and negative %ME were interpreted as matrix suppression, whereas MF values > 1 and positive %ME were interpreted as an enhancement. The assay was also performed to ascertain specificity and create a heatmap to demonstrate the cross-reactivity percentage using Equation (5), adapted from a previous publication [75].
Cross-reactivity% = 100 × (OD with target analyte)/(OD with cross reactant analyte)(5)

In Equation (5), OD refers to the optical density value at 450 nm with target or cross-reactant mycotoxins at a concentration that reduces the detection signal by at least 50%.

## Figures and Tables

**Figure 1 toxins-14-00727-f001:**
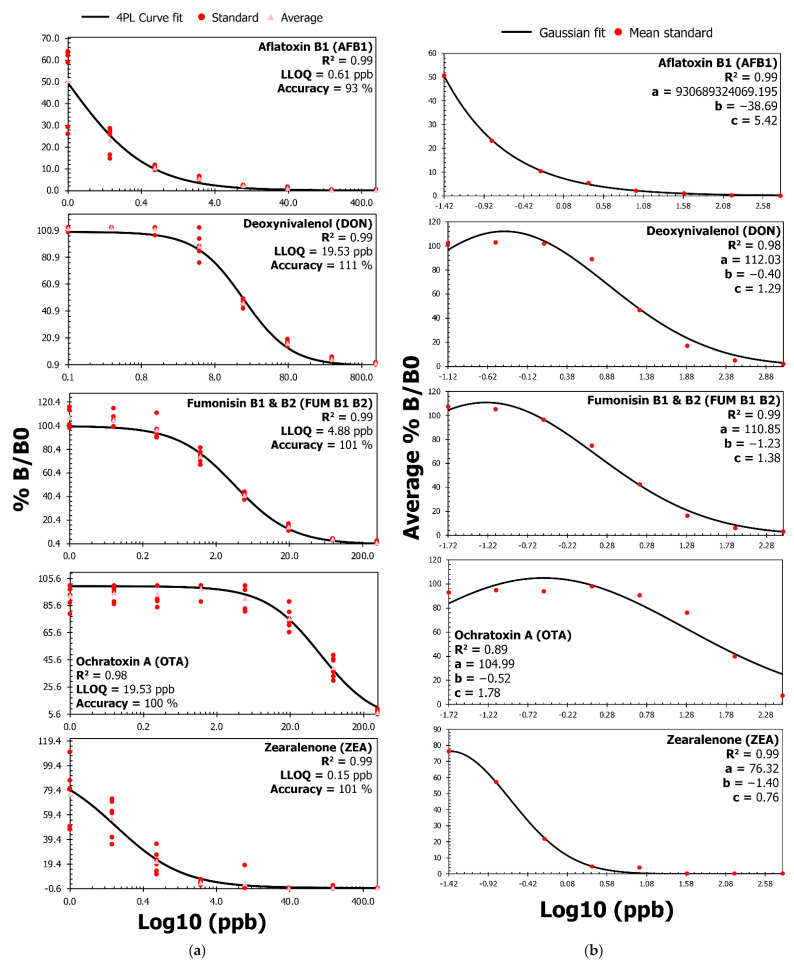
Calibration curves with a 4PL and Gaussian fit for AFB1, DON, FUM, OTA, and ZEA. (**a**) The assay can accurately measure low amounts of analyte as the four-parameter logistic (4PL) curve establishes a near-perfect fit (R²) for all mycotoxins. The coefficient of determination (R²) is the ratio of the variation that the 4PL curve-fitting model explains to the total variation in the model. The R² value will equal 1 for a perfect fit and will be closer to 0 for a bad 4PL fit. Furthermore, the lower limit of quantitation (LLOQ) is the lowest amount of an analyte detectable in a sample with suitable accuracy. As a result, Figure 1a presents the LLOQ in ppb for each mycotoxin and the associated accuracy percentages at the LLOQs achieved through back-calculation. (**b**) Gaussian curve fit analysis and associated coefficient values used to predict the concentration (in ng/mL or ppb) for recovery and the matrix interference analysis are in Table 1. The a, b, and c Gaussian coefficients refer to the height of the curve’s peak, the position of the center of the peak, and the standard deviation, respectively. In panels a and b, %B/B0 refers to the OD value with (B) and without (B0) competition for antibody binding sites on the microplate.

**Figure 2 toxins-14-00727-f002:**
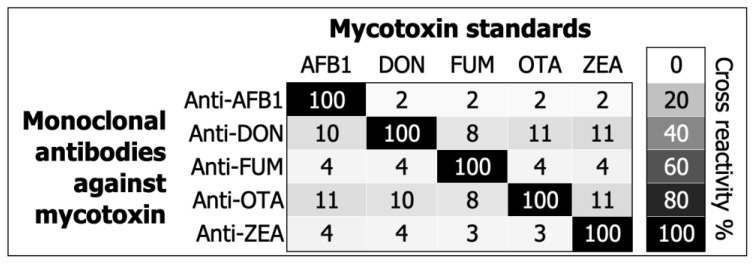
Monoclonal antibodies against specific mycotoxins do not significantly cross-react.

**Table 1 toxins-14-00727-t001:** Mycotoxin extraction from human sera results in repeatable (CV%) recovery percentages for all analytes. The recovery percentage refers to the log10 concentration (ng/mL or ppb) of analyte recovered from spiked samples before versus after extraction. Matrix effect (ME) and matrix factor (MF) help to determine the influence of human sera on recovery percentages. Ideally, MF will equal 1, and ME should be roughly 0% to indicate that the extraction process is neither suppressing nor enhancing the recovery results.

Mycotoxin	Spike (ppb)	Mean Recovery (%)	Mean Repeatability (CV%)	Mean Matrix Effect (%)	Mean Matrix Factor
AFB1	0.61	96	0.05	−72.11	0.27
	2.44	97	4	−65.91	0.33
	9.77	96	4	−49.71	0.48
	39.06	101	2	−40.50	0.60
DON	19.53	98	4	−9.28	0.88
	78.13	94	5	−0.78	0.93
	312.50	94	7	6.19	1.00
	1250	91	17	8.75	0.99
FUM B1/B2	4.88	73	0.1	5.47	0.77
	19.53	81	11	−0.44	0.80
	78.13	79	6	3.12	0.81
	312.50	91	2	−3.76	0.88
OTA	19.53	81	0.4	−9.51	0.74
	78.13	86	21	−10.16	0.76
	312.50	79	3	−3.69	0.76
	1250	92	6	−7.43	0.85
ZEA	0.15	85	4	−39.22	0.52
	0.61	95	3	−29.88	0.66
	2.44	89	10	−12.69	0.78
	9.77	106	18	−10.12	0.94

**Table 2 toxins-14-00727-t002:** The LLOQ from the present study is amongst the lowest values observed in commercial and emerging immuno-based methods. The reference tests in the present table were not optimized for mycotoxin detection in human serum (Appendix A). As a result, the performance of the assays compared in Table 2 should not be considered a head-to-head comparison but a comparison between detection techniques that have employed similar formats for establishing detection (i.e., direct detection) and calibration curves (i.e., non-linear) for Aflatoxin B1 (AFB1), deoxynivalenol (DON), fumonisin (FUM), ochratoxin A (OTA), and zearalenone (ZEA).

Parameters or Studies for Comparison	AFB1 (ppb)	DON (ppb)	FUM (ppb)	OTA (ppb)	ZEA (ppb)
Lowest EU guidance levels for food ^°^	2 ^◊^	500 ^◊^	800 ^◊^	2 ^◊^	50 ^◊^
LLOQ from the present study *	0.61	19.53	4.88	19.53	0.15
Cusabio (ELISA) ^§^	1.5	100	30	1.5	30
Elabscience (ELISA) ^⊗^	0.6	150	20	5	6
Helica^TM^ (ELISA) ^⊕^	4	500	100	1	NI
AgraQuant^®^ (ELISA) ^#^	2	250	250	2	25
VICAM (LFIA) ^^^	2	250	200	2.5	100
Wu et al. 2020 (LFIA) ^∅^	0.1	NA	4	0.2	0.8
Xing et al. 2020 (LFIA)	4	200	20	NA	40
Charlermroj et al. 2021 (LFIA) ^∅^	5	10	0.5	NA	10
Joshi et al. 2016 (SPR) ^∅^	3	26	10	13	16
Wie et al. 2019 (SPR) ^∅^	0.9	5.3	NA	1.9	10.3

**^°^** Excluding foodstuffs for infants and young children. **^◊^** Food commodities with the lowest guidance levels for mycotoxins are as follows: AFB1 = groundnuts, tree nuts, dried fruit, and all cereals, including maize and rice; DON = bread, biscuits, pastries, cereal snacks, and breakfast cereals; FUM = maize-based breakfast cereals and maize-based snacks; OTA = flavored or fruit wine and grape juice or its concentrate; ZEA = bread, pastries, biscuits, cereal snacks, and breakfast cereals. ***** LLOQ presented in ppb because 1 ng/mL = 1 ppb. Refer to Figure 1 in the results section. **^§^** Lowest detection range is indicated on the website [38]. **^⊗^** Detection limits were noted from the Elabscience website for AFB1, DON, FUM, OTA, and ZEA [40,41,42,43,44]. **^⊕^** Detection limits were noted from the Hygiena website for AFB1, DON, FUM, OTA, and ZEA [45,46,47,48,49]. **^#^** Limit of quantitation noted from the AgraQuant^®^ website for AFB1, DON, FUM, OTA, and ZEA [50,51,52,53,54]. **^^^** Limit of detection noted from the VICAM website for AFB1, DON, FUM, OTA, and ZEA [55,56,57,58,59]. **^∅^** Lowest ppb in the linear range as indicated by the authors [33,34,35,36,37]. ELISA = enzyme-linked immunosorbent assay; LFIA = lateral flow immunoassay; SPR = surface plasmon resonance; NA = not applicable; NI = not indicated.

**Table 3 toxins-14-00727-t003:** The recovery ranges in the present study, commercial tests, and emerging immuno-based methods are comparable despite differences in the extraction protocols and matrices. The reference tests in the present table were not optimized for mycotoxin detection in human serum (Appendix A). As a result, the performance of the assays compared in Table 3 should not be considered a head-to-head comparison but a comparison between detection techniques that have employed similar formats for establishing detection (i.e., direct detection) and calibration curves (i.e., non-linear) for Aflatoxin B1 (AFB1), deoxynivalenol (DON), fumonisin (FUM), ochratoxin A (OTA), and zearalenone (ZEA).

Parameters or Studies for Comparison	Recovery (%)				
	AFB1	DON	FUM	OTA	ZEA
Present study	96–101	91–98	73–91	79–92	85–106
Elabscience (ELISA)	69–99	70–100	78–108	70–100	70–100
Hygiena Helica^TM^ (ELISA)	82–109	74–82	82–119	95–101	89–102
Wu et al. 2020 (LFIA)	85–112	NA	88–112	82–116	88–104
Charlermroj et al. 2021 (LFIA)	87–111	87–109	88–108	NA	89–123
Wie et al. 2019 (SPR)	92–104	88–104	NA	95–111	89–103

NA = not applicable. Recovery values were noted from the Elabscience and Hygiena websites for AFB1, DON, FUM, OTA, or ZEA [40,41,42,43,44,45,46,47,48,49] and other publications [33,35,37,43,45].

## Data Availability

All data is presented in the figures, tables, and Appendix A of the article.

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
