# Peer review of "Analytical Validation of a Direct Competitive ELISA for Multiple Mycotoxin Detection in Human Serum"

_toxins, 2022, doi:10.3390/toxins14110727_

Round 1

Reviewer 1 Report

This paper reminds the problematic of the dosage of mycotoxins in fluids and tissues and the difficulty to do so and to correlate the results with the values measured in food and feed.

The development of this rapid method of analysis is interesting and could be a nice tool in the field.

I have some questions/comments :

 -In the abstract and in the key contribution, it's not precised which fumonisin is measured.

-L44 : when you sum up in one sentence the action of each mycotoxin, it’s strange that you cite this one in particular for the DON. Beacuse it’s not the one that we think when we talk about DON.

-Table 2 and table 3 : can you indicate for which matrices are these results, because it’s not clear in which matrices are done each test that you compare.

- In the legend of figure 1, « %B/B0 » is not defined and there are « no b) »

Globally, improvement of the captions can be done for all the table/figures.

-Concerning the validation of your method, why didn’t you do a comparison with a reference method, as HPLC-UV or HPLC-MS, to confirm your results and your sensitivity of detection ?

-In the discussion part you don’t discuss the problematic of correlating your results to what it’s found in raw materials/feed/food ; Do they exist a correlation factor and for each mycotoxin ?

Because it’s really an important problematic, how to use your obtained data in sera and correlate them to the mycotoxins exposure and to the toxic effects known.

Author Response

Please see attachment for point by point response and actions to your review. With regards 

Reviewer 2 Report

The authors developed a direct competitive ELISA to facilitate the detection of aflatoxin B1 (AFB1), deoxynivalenol (DON), fumonisin (FUM), ochratoxin A (OTA), and zearalenone (ZEA) in human serum. The authors employed analytical validation that followed practices endorsed by the international research community and EU directive 96/23/EC to examine detection capability, recovery, and cross-reactivity. Interesting results were reported, however, there are some areas that can be improved:
a) The introduction seems insufficient. Please expand introduction to capture more details regards various analytical methods, then narrow it down to direct enzyme-linked immunosorbent assay (ELISA), and tell us why this one is unique. This should be a paragraph on its own. Next, tell us about the challenges in mycotoxin detection, and what researchers have done about it, this should be another paragraph. Authos, use your discretion on how you desire to reorganise the introduction. However, these suggestions need to feature in the introduction. Please, strengthen your rationale why this study is required.
b) Results are ok. In the discussion, please kindly extract better the specific results that are being discussed, so that readers can clearly link which is being discussed and why. Also, all Tables and Figures in the results must be mentioned in the discussion, using "(Refer to Table x)" or "(Refer to Figure x)"

It seems your numbering went from 3 to 5, did you omit the conclusions?
The reviewer suspects so, therefore please include your conclusion in your revised manuscript, and make sure it ties up the work well, and kindly include recommendation for future work.
A very promising work,

Author Response

(The authors gave the same response as above.)

Reviewer 3 Report

General comment: The biological validity of determining free mycotoxins in serum as exposure biomarkers should be discussed when the majority fraction that appears in serum is mostly mycotoxin metabolites (not intact toxins).

Abstract: the term ppb should be avoided, use µg/kg or µg/L instead

Results: According to the title, the manuscript deals with an analytical validation. In general, bioanalytical methods based on immuno-recognition or receptor binding are (generally) semi-quantitative screening methods. Therefore, positive (and negative) results should be confirmed by instrumental techniques, at least for method validation. This exercise has not been performed by the authors.

There should be a short introductory sentence to present the results so that the reader knows what the authors are talking about.

In general, the different validation parameters (e.g. linearity, sensitivity, recovery, precision, etc.) should be dealt with in separate paragraphs for clarity.

All abbreviations should be defined the first time they appear in the manuscript (e.g. 4PL, OD, etc.)

Line 76, 85: use ‘showed’ (or similar) instead of ‘demonstrated’

Figure 1: the caption contains explanations that could be moved to the text

Table 1: the title contains explanations that could be moved to the text

Table 2: the table 2 compares present method with several commercial immune-based methods. The question is, are these commercial methods intended to analyze mycotoxins in human serum or in food and feed? If the methods are only intended for food and feed, the comparison of performance is not adequate as the methods are validated for other commodities. Although this limitation is mentioned in the Discussion, I believe that the Table 2 is not appropriate and should be corrected.

Table 2: the captions contains a lot of information. Most explanations could be moved to the text.

Line 116: Cross-reactivity between structurally unrelated mycotoxins is of less interest than cross-reactivity between structurally related mycotoxins (e.g. the four aflatoxins, the 3- and 15-acetylated forms of DON, the OTAs, the ZEAs, etc.) or between free mycotoxins and their serum metabolites that are expected to be present in real samples.

Line 158-159: The EU does not have mycotoxin ‘recommended levels’ but ‘maximum levels’.

Material and methods:

Negative control (blank matrix) samples are missing, that is a sample known to be free of the mycotoxin to be screened for, e.g. by previous determination using a confirmatory method of sufficient sensitivity.

Other limitations of the study:

1/ The result of the new ELISA test when applied to real serum samples is assumed to be a numerical value. Then, the new method should have been tested with real human serum samples of unknown mycotoxin concentrations and validated against instrumental methods such as LC-DAD-FLD or LC-MS for comparison of results. Perhaps, results from positive and negative samples should have been verified by a full re-analysis from the original sample by a confirmatory method.

2/ For the validation procedure, the false negative rate and false suspected rate should be determined with the appropriate samples and cut-off values.

Author Response

(The authors gave the same response as above.)

Round 2

Reviewer 2 Report

Thank you authors for revising your work, and for addressing the concerns raised. The reviewer is satisfied with the revised manuscript and believes it is acceptable for publication.

Reviewer 3 Report

In my opinion, the authors' answers to the questions raised are sufficient. The manuscript has been corrected and improved making clear the limitations of the study and the future work to be done. With these considerations, I believe that the current revised version is publishable in Toxins.